# Trehalose Modulates Autophagy Process to Counteract Gliadin Cytotoxicity in an In Vitro Celiac Disease Model

**DOI:** 10.3390/cells8040348

**Published:** 2019-04-12

**Authors:** Federico Manai, Alberto Azzalin, Martina Morandi, Veronica Riccardi, Lisa Zanoletti, Marco Dei Giudici, Fabio Gabriele, Carolina Martinelli, Mauro Bozzola, Sergio Comincini

**Affiliations:** 1Department of Biology and Biotechnology, University of Pavia, 27100 Pavia, Italy; alberto.azzalin@unipv.it (A.A.); martina.morandi01@universitadipavia.it (M.M.); veronica.riccardi02@universitadipavia.it (V.R.); lisa.zanoletti01@universitadipavia.it (L.Z.); marco.deigiudici01@universitadipavia.it (M.D.G.); fabio.gabriele01@universitadipavia.it (F.G.); carolina.martinelli01@universitadipavia.it (C.M.); sergio.comincini@unipv.it (S.C.); 2Pediatrics and Adolescentology Units, Department of Internal Medicine and Therapeutics, University of Pavia, 27100 Pavia, Italy; mauro.bozzola@unipv.it

**Keywords:** celiac disease, gluten, gliadin, autophagy, Caco-2 cells, trehalose

## Abstract

Celiac disease (CD) is a chronic systemic autoimmune disorder that is triggered by the ingestion of gliadin peptides, the alcohol-soluble fraction of wheat gluten. These peptides, which play a key role in the immune response that underlies CD, spontaneously form aggregates and exert a direct toxic action on cells due to the increase in the reactive oxygen species (ROS) levels. Furthermore, peptic-tryptic digested gliadin peptides (PT-gliadin) lead to an impairment in the autophagy pathway in an in vitro model based on Caco-2 cells. Considering these premises, in this study we have analyzed different mTOR-independent inducers, reporting that the disaccharide trehalose, a mTOR-independent autophagy activator, rescued the autophagy flux in Caco-2 cells treated with digested gliadin, as well as improved cell viability. Moreover, trehalose administration to Caco-2 cells in presence of digested gliadin reduced the intracellular levels of these toxic peptides. Altogether, these results showed the beneficial effects of trehalose in a CD in vitro model as well as underlining autophagy as a molecular pathway whose modulation might be promising in counteracting PT-gliadin cytotoxicity.

## 1. Introduction

Celiac disease (CD) is a chronic systemic autoimmune disease belonging to the gluten-related disorders (GRDs), a spectrum of diseases triggered by the ingestion of gluten, which affects the small intestine of individuals with a genetic predisposition [1]. Gluten is a composite of storage proteins, present in wheat, barley, rye, oats, and related species. Specifically, the pathological immune response is caused by the alcohol soluble fraction of gluten, i.e., gliadin, which shows toxic properties especially after the enzymatic digestion mediated by pepsin and trypsin [2]. Among the four different types of gliadin, i.e., ω5-, ω1-,2-, α/β-, and γ- gliadin [3,4], the most biologically active peptide, which exerts cytotoxic activities and immunological reactions, derived from α-gliadin [5]. Particularly, the intracellular accumulation of the gliadin peptide p31-43 leads to an increase in reactive oxygen species (ROS) levels [6], and the highly immunogenic 33-mer peptide is able to form supramolecular organized immunoreactive structures resistant to low pH [7,8,9]. Although autophagy dysregulation has been largely described in different autoimmune disorders affecting the intestine, such as inflammatory bowel diseases (IBDs) [10,11], little evidence was collected in our laboratory supporting the hypothesis that autophagy could play a role in CD pathogenesis [12,13]. A major reason to thoroughly investigate autophagy in the context of CD is that this cellular process is implicated in the oxidative stress response [14] and in protein aggregates degradation [15], as well as in the maintenance of the barrier function in the epithelial cells of the small intestine [16]. Moreover, mutations in TBC1D7, a protein that participates in the formation of the TSC1/2 complex, were described in celiac patients affected by familiar syndromes with a complex clinical presentation [17]. In actual fact, the only effective treatment of CD is a gluten-free diet (GFD), although it shows limitations and disadvantages [18]. According to all these considerations, autophagy modulation seems to be an interesting approach in order to counteract gliadin cytotoxicity. As already demonstrated, autophagy induction through different approaches that inhibit mTOR showed poor results as compared with other ones acting on different kinases, for example, 3-methyladenine [12], which suggests the need to test novel autophagy modulators. Recently, it has been highlighted that trehalose, raffinose, and sucrose are able to induce autophagy through an mTOR-independent mechanism [19]. In particular, the natural disaccharide, trehalose, alleviates inflammation and oxidative stress response [20], as well as prevents the aggregation of toxic peptides in pathological conditions, such as Huntington disease [21]. The working hypothesis is that trehalose, an mTOR-independent autophagy inducer, might be able to activate the autophagy catabolic process, thus leading to a reduction in the cytotoxicity of PT-gliadin resulting in an improvement of cell viability.

Considering these premises, in this study we investigated through different techniques the effect of trehalose on counteracting gliadin cytotoxicity in the context of CD. In particular, we assessed the capability of trehalose to induce autophagy in a CD in vitro model based on the full confluent Caco-2 cells monolayer, and finally to improve cell viability.

## 2. Materials and Methods

### 2.1. Cell Culture and Autophagy Modulation

The Caco-2 and HT-29 cells from the American Type Culture Collection (ATCC) were cultured at 37 °C in a 5% CO_2_/95% atmosphere and maintained in a DMEM medium (Euroclone, Milano, Italy), supplemented with 10% FBS, 100 units/ml penicillin, 0.1 mg/ml streptomycin, and 1% L-glutamine. The Caco-2 and HT-29 cells (2.5 × 10^5^) were seeded in a 6-well culture plate and analyzed once they reached a full complete confluence (3 × 10^6^). Autophagy was induced through an incubation of 24 h with different inducers: rapamycin (5 µM); nicotinamide (5 mM); metformin (5 mM); trehalose, raffinose, and sucrose (100 mM) (Sigma-Aldrich, St. Louis, MO, USA); and SMER-28 (50 µM) (Selleck Chemicals, Munich, Germany). The culture medium was replaced with fresh DMEM 24 h before every treatment to avoid the risk of nutrient deprivation. Bafilomycin A1 (10 nM) was administered 6 h before cells collection to evaluate the autophagy flux. 

### 2.2. Gliadin Digestion and Fluorescent Labelling

Gliadin from wheat (Sigma) was digested as described by Drago et al. with minor modifications [22]. Specifically, gliadin (1 g/mL) was resuspended in 500 mL 0.2 N HCl for 2 h at 37 °C with 1 g pepsin (Sigma). Then, gliadin was further digested by the addition of 1 g of trypsin (Sigma) after the pH was raised to 7.4 using 2 N NaOH, and the solution was incubated at 37 °C for 4 h under agitation. Finally, enzymes were inactivated by boiling the solution for 30 min and the preparation (referred to as PT-gliadin) was stored at −80 °C. Albumin and casein (Sigma) were digested following the same protocol and were used as negative controls. PT-gliadin AlexaFluor555 labelling (GLIA-555) and purification was performed, as described previously [12], using affinity chromatography-purification G50 column (GE Healthcare, Little Chalfont, UK). The resulting labelled proteins were resuspended in PBS and labelled according to the manufacturer’s instructions with AlexaFluor555 using Alexa Fluor Microscale Labelling Kits (ThermoFisher, Waltham, MA, USA). The intracellular content of GLIA-555 was evaluated through cytofluorimetric analysis using a Muse Cell Analyzer (Merck, Darmstadt, Germany). The digested gliadin, casein, albumin, and GLIA-555 were administered at a final concentration of 1 µg/µL for 24 h. 

### 2.3. Immunoblotting Analysis

Caco-2 cells were cultured in a 6-multiwell plate in presence/absence of digested gliadin, casein or albumin (1 µg/µL), as described before. Immunoblotting was performed as described [12]. Cells were collected and lysated in ice-cold Triton X-100 (50 mM Tris-HCl pH 7.4, 150 mM NaCl, 1% Triton X-100) supplemented with Complete Mini protease inhibitor cocktail 7X (Roche, Basel, Switzerland) and sodium orthovanadate 1 mM (Sigma). Proteins were quantified using the Quant-It Protein Assay Kit (Invitrogen, Carlsbad, CA, USA). Proteins (20–30 µg) were added to a Laemmli sample buffer (2% SDS, 6% glycerol, 150 mM β-mercaptoethanol, 0.02% bromophenol blue, and 0.5 M Tris-HCl, pH 6.8), denaturated for 5 minutes at 95 °C and separated on 12% SDS-PAGE according to protein size. After electrophoresis, proteins were transferred onto a nitrocellulose membrane using the Trans-Blot Turbo Transfer System (Biorad, Hercules, CA, USA) according to the manufacturer’s instructions. Then, the membranes were blocked for 1 hour at room temperature (RT) with 5% (*w/v*) non-fat milk in TBS (138 mM NaCl, 20 mM Tris-HCl, pH 7.6) containing 0.1% Tween-20, and incubated overnight at 4 °C with polyclonal primary antibodies LC3-II, ATG5, Beclin1, and p62 (Cell Signaling, Danvers, MA, USA) diluted at 1:2000 in 5% non-fat milk in TBS, whereas monoclonal primary antibody against β-actin (BACT) was diluted at 1:5000 (Cell Signaling). Species-specific peroxidase-labelled ECL secondary antibodies (Cell Signaling, 1:2000 dilution) were used in 5% non-fat milk in TBS. Proteins signals were detected using the ECL Prime Western Blotting Detection Kit (GE Healthcare) by means of a Chemidoc MP system (Biorad). The densitometric analysis was conducted with ImageJ software (http://rsbweb.nih.gov/ij).

### 2.4. Apoptosis and Autophagy Cytofluorimetric Analysis

The viability and apoptotic rate of Caco-2 cells were quantified using the Muse Annexin V and Dead Cell Assay (Merck), according to the manufacturer’s instructions. After trypsinization and collection, Caco-2 cells were washed 3 times with PBS, resuspended in PBS + 1% FBS (*v/v*) and 1 volume of Annexin V reagent, and incubated for 20 min at RT in the dark. The analysis was performed using a Muse Cell Analyzer (Merck). Autophagy detection was assayed using the Autophagy LC3 Antibody-based Kit (Merck). Specifically, Caco-2 cells were permeabilized and then incubated in ice for 30 min with the anti-LC3 mouse monoclonal AlexaFluor555 conjugated antibody, according to the manufacturer’s instructions. The analysis was performed using the Muse Cell Analyzer (Merck). 

### 2.5. Multispectral Imaging Flow Cytometry (MIFC) and Spot-Count Analysis. 

After trypsinization, Caco-2 cells were fixed using 4% of paraformaldehyde for 15 min at RT. Then, LC3-II labelling was performed using the Autophagy LC3 Antibody-based Kit (Merck) according to the manufacturer’s instructions in order to validate the results obtained using the Muse Cell Analyzer (Merck). Finally, 1 × 10^6^ cells were resuspended in 200 µL of D-PBS for the analysis and 2000 events were collected for every sample. Spot-count analysis was performed using the spot-count feature Spot Count_Range(Peak(Spot(M03,Ch3-LC3-AF555,Bright,7,3),Ch3,Bright,0). 

### 2.6. Statistical Analysis

The data were analyzed using the statistics functions of the MedCalc statistical software version 18.11.6. (http://www.medcalc.org). The Anova test differences were considered statistically significant when *p* ≤ 0.05. 

## 3. Results

### 3.1. PT-Gliadin Administration Leads to Autophagy Blockage and Cell Death in Caco-2 Cells Monolayer

A schematic representation of the experimental plan was followed to study the effects of digested gliadin on autophagy in an in vitro model based on full confluent Caco-2 cells and the beneficial effects of trehalose is shown in Table 1.

Preliminarily, autophagy levels were investigated in subconfluent versus full confluent Caco-2 cells monolayer. Specifically, LC3-II levels were analyzed through immunoblotting, as seen in Figure 1, showing a marked increase at full confluence cell density. 

Subsequently, the autophagy response in a Caco-2 cells monolayer at full confluence following PT-gliadin (GL) administration was evaluated. The Caco-2 cells were cultured for 5 days after they reached complete monolayer confluence and then were treated with digested gliadin as described in the material and methods section. Cytofluorimetric analysis of LC3-II levels was assayed at different time intervals (i.e., 6, 24, and 48 h post-treatment, p.t.). As shown in Figure 2, no statistically significant differences were detected in LC3-II expression levels between Caco-2 treated with PT-gliadin and untreated (NT) cells. As expected, LC3-II levels increased after bafilomycin A1 administration, mostly at 24 h p.t., in NT cells as compared with those treated with the digested gliadin peptides. 

Then, immunoblotting analyses of LC3-II and p62 protein expression were performed to thoroughly investigate the autophagy response following PT-gliadin administration. Similar to the cytofluorimetric analysis, no statistically significant differences in LC3-II expression levels were detected between NT sample and Caco-2 cells treated with PT-gliadin (Figure 3A,B). However, in this case, an increase in LC3-II levels was observed in NT cells treated with bafilomycin A1. A similar trend was observed for the expression levels of p62 (Figure 3C).

A similar experiment was performed in presence of PT-casein and PT-albumin (i.e., PT-CAS and PT-ALB), respectively. The LC3-II levels were analyzed after 24 h p.t. by means of cytofluorimetric analysis and immunoblotting. As summarized (Figure 4), no statistically significant differences were detected between the samples treated with digested casein (CAS) and albumin (ALB) as compared with NT samples. On the contrary, significant differences in LC3-II expression levels were scored in the samples in presence of bafilomycin A1 as compared with both their relative controls and NT samples. 

In addition, the experimental set was analyzed through immunoblotting and densitometric analyses (Figure 5). According to the previous results, an increase in LC3-II expression levels was detected in the samples treated with bafilomycin A1 as compared the relative negative controls and the NT samples.

Finally, cell viability in presence of PT-gliadin was assayed at 24 h p.t. through Annexin V cytofluorimetric analysis as compared with untreated cells. As reported in Figure 3, a statistically significant decrease in cell viability was observed after PT-gliadin administration as compared with the NT sample but not in presence of digested casein and albumin proteins. Furthermore, GL administration induced a marked increase in apoptotic/dead index as compared with other treatments. 

### 3.2. Single Cell Analysis Confirms Caco-2 Autophagy Blockage After PT-Gliadin Administration

In order to confirm the effects exerted on the autophagy process by digested gliadin in Caco-2 cells monolayer, a multispectral imaging flow cytometry (MIFC) analysis was performed to evaluate induced morphological and autophagy differences. First, as reported in Figure 4, in order to detect possible differences, the cell diameters of the Caco-2 cells treated/untreated with enzymatically digested gliadin were determined. No statistically significant differences were observed between the two groups. Following LC3-II staining, a spot counting analysis was subsequently performed, as described in the methods. As shown in Figure 6, a significant increase in LC3-II spots in untreated (NT) Caco-2 cells with bafilomycin A1 as compared with the NT sample was reported. Furthermore, no significant increase was scored in PT-gliadin and bafilomycin A1 treated cells as compared with the relative control (GL). 

### 3.3. Trehalose, an mTOR-Independent Inducer of Autophagy, Increases Autophagic Flux and Counteract PT-Gliadin Cytotoxicity Reducing its Intracellular Content

As already reported, rapamycin, a well-known mTOR-dependent autophagy activator, did not produce a significant improvement in the clearance of extra- and intracellular fluorescent PT-gliadin amount [12]. Consequently, different mTOR-independent autophagy inducers, i.e., trehalose, raffinose, and sucrose (all at 100 mM), as well as SMER-28 (50 µM), were tested in Caco-2 cells monolayer for their capability to counteract gliadin toxicity through autophagy, and these were compared with other autophagy inducers such as rapamycin (5 µM), metformin (5 mM), and nicotinamide (NAM, 5 mM). The LC3-II expression levels were evaluated after 24 h p.t. by means of cytofluorimetric analysis. As reported in Figure 7, a statistically significant increase in LC3-II levels was observed in Caco-2 cells treated with trehalose and raffinose, while rapamycin, metformin, SMER-28, NAM, and sucrose did not produce significant variations. In the same panel, chloroquine (25 µM) and bafilomycin A1 (10 nM) were assayed to evaluate the autophagic flux. 

Then, the effect of saccharides (i.e., trehalose and raffinose) on Caco-2 cell viability was investigated through Annexin V cytofluorimetric analysis. As reported in Table 2, no statistically significant differences were detected in the ratio of the treatments as compared with untreated cells. 

Subsequently, an immunoblotting survey plot of the main autophagy-related proteins (i.e., ATG5, Beclin1, and LC3-II) was performed (Figure 8), reporting that among the investigated saccharides, trehalose produced the highest autophagy response in proteins expression. 

The LC3-II expression levels were then analyzed in order to investigate the ability of trehalose and raffinose to rescue autophagy in presence of PT-gliadin. The Caco-2 cells were incubated for 24 h with the digested gliadin, trehalose, and with the combination of both. The LC3-II expression profiles were examined by cytofluorimetric and immunoblotting assays. As reported in Figure 9, trehalose treatment induced a significant increase of LC3-II expression in presence of digested gliadin, as highlighted in the autophagy flux differences. 

The autophagic flux rescue mediated by trehalose was also investigated through the analysis of LC3-II and p62 expression by immunoblotting. The results reported in Figure 10, indicate that in presence/absence of bafilomycin A1 (10 nM), the treatments with trehalose and trehalose combined with gliadin determined a statistically significant increase of LC3-II and p62 levels as compared with the untreated samples.

The ability of trehalose to induce autophagy in presence/absence of digested gliadin was also investigated through spot counting analysis performed by MIFC. As reported in Figure 11, a significant increase in LC3-II positive spots was observed in presence of trehalose as compared with the untreated (NT) Caco-2 cells. Similarly, there was a marked increase scored in cells treated with both trehalose and PT-gliadin.

Then, Annexin V cytofluorimetric analysis was performed in order to evaluate whether autophagy activation mediated by trehalose affected viability and apoptotic cell death. As summarized in Figure 12, trehalose administration in presence of PT-gliadin, following the same experimental scheme as described above, rescued cell viability, reducing the apoptotic rate induced by digested gliadin treatment. 

Finally, the ability of trehalose to reduce the intracellular amount of PT-gliadin was evaluated, as previously described [12]. The Caco-2 cells were incubated for 24 h with GLIA-555, trehalose, and with the combination of both. Intracellular fluorescence of GLIA-555 was evaluated by means of cytofluorimetric assay (exc. 555 nm, em. 580 nm). As reported below (Figure 13), the intracellular fluorescence content was significantly reduced in Caco-2 cells treated with trehalose as compared with those treated with GLIA-555. In the control untreated sample (NT) and in trehalose treated cells no significant fluorescence signals were detected. 

### 3.4. Trehalose Activates Autophagy in HT-29 Cells Following PT-Gliadin Administration

The effect of trehalose in modulating the autophagy process in response to digested gliadin administration was also evaluated in human HT-29 undifferentiated colorectal cancer cells. Preliminarily, viability and cell death assays were performed through Annexin V cytofluorimetric analysis, comparing the effect of digested gliadin and casein proteins with the untreated samples. As reported in Table 3, statistically significant variations in cell viability and apoptotic death were only scored following PT-gliadin administration. Furthermore, cells treated with PT-casein did not exhibit any differences in their viability/apoptotic rates as compared with untreated samples. 

Then, PT-gliadin and trehalose were administered for 24 h to HT-29 cells, as single treatments or combined ones, in order to evaluate the autophagic response to gliadin and the beneficial effects exerted by trehalose. The LC3-II expression levels were assayed using cytofluorimetric assays. As reported in Figure 14, PT-gliadin administration increased LC3-II levels in HT-29 cells as compared with the untreated sample. Similar to the results obtained with Caco-2 cells, no statistically significant increase of LC3-II expression was detected following PT-gliadin and bafilomycin A1 combined administration (GL vs GL + BAF).

## 4. Discussion

In this work, in vitro experiments were conducted to investigate the autophagy response to enzymatically digested gliadin in a model based on the full confluent Caco-2 cells monolayer, and the beneficial effects of trehalose administration to counteract the cytotoxicity of gliadin peptides. 

As we described previously, the administration of peptic-tryptic digested gliadin decreased autophagy in Caco-2 cells, as well as established an in vitro model for polarized epithelial transport and CD due to its physiological similarities with small bowel enterocytes [24,25], thus promoting cell death through apoptosis [12]. Considering that high cell density suppresses mTOR in both normal and cancer cells, and consequently, autophagy levels increase in Caco-2 cells according to their confluence and differentiation state [26,27], in vitro experiments were performed in Caco-2 cells monolayer exhibiting differences in cell density. Preliminarily, we confirmed through immunoblotting analysis that Caco-2 cell density affects the basal autophagy levels. Then, Caco-2 cells were cultured until the formation of a complete confluent monolayer in order to investigate the autophagy response after PT-gliadin administration. As evidenced by cytofluorimetric assay and immunoblotting analysis of LC3-II and p62 proteins, autophagy blockage and an increase in the apoptotic rate were observed at different time intervals, underlining that gliadin peptides exert their cytotoxic effects also in high density culture conditions. On the other hand, enzymatically digested albumin and casein were tested as controls in the same experimental conditions and no toxicity was detected. In addition, the autophagy blockage observed after PT-gliadin administration was evaluated by the multispectral imaging flow cytometry (MIFC) approach in order to study autophagy at the single cell level. This technique as compared with standard flow cytometry (FC) is an emerging field of imaging technology, which combines both fluorescence/confocal microscopy and flow cytometry techniques. The advantage of MIFC is that it has the ability to make quantitative morphological and spatial measurements from a large population of single cells [28]. By performing a spot-count analysis, the impairment of the autophagy flux in Caco-2 cells was demonstrated, as evidenced by the reduced accumulation of LC3-positive puncta following PT-gliadin and bafilomycin A1 treatments, as compared with their negative control. 

Autophagy is triggered through different mechanisms by a variety of stimuli, i.e., protein aggregates and oxidative stress [14,29]. However, in the presence of persistent stresses and inadequate autophagy response, cells activate the apoptotic pathway leading to cell death [30]. Gliadin behavior resembles those of other toxic peptides that form aggregates, such as β-amyloid and α-synuclein. As previously stated, autophagy activation could represent a valid therapeutic opportunity in neurodegenerative disorders caused by the accumulation of toxic proteins [31,32]. In a celiac context, the observed autophagy blockage is explained by the ability of the immunodominant 33-mer peptide to spontaneously form supramolecular structures, whose degradation is difficult, and the increase of ROS caused by the internalization and the accumulation of the p31-43 peptide [6,7,33].

Considering this evidence, autophagy induction seems to be a promising approach to counteract the toxic effects of gliadin peptides, as we suggested [12,13]. This strategy is strengthened by the key role that autophagy plays in the maintenance of intestinal homeostasis [34]. A decrease in the levels of proteins involved in tight junctions (TJs) formation has been described as a main step in the pathogenesis of inflammatory diseases of the intestine, such as Crohn disease, which are characterized by the loss of the selective permeability properties of the intestine. Autophagy promotes the maintenance of the correct levels of TJs through the degradation of claudin-2, which is responsible for the formation of cation-selective pores. In addition, autophagy impairment increases levels of TNF-α which leads to an increase in the claudin-2 levels and, as consequence, to a loss in the intestinal barrier properties [16]. Other evidence of the beneficial effects of autophagy induction in inflammatory intestinal diseases has been obtained [35], demonstrating that autophagy induction through betanin, rapamycin, and trehalose improves the clinical presentation in a murine model of inflammatory bowel diseases (IBDs). In our previous work [12], we have investigated the efficacy of three molecular approaches to induce autophagy in presence of digested gliadin, i.e., starvation, rapamycin, and 3-metyladenine (3-MA). Unfortunately, among these treatments, only 3-MA showed encouraging results, whereas rapamycin, a well-known mTOR inhibitor [36], did not show a marked efficacy. For these reasons, we tested additional autophagy modulators, i.e., NAM, metformin, SMER-28, trehalose, raffinose, and sucrose, whose mechanisms of induction are mTOR-independent. NAM, as well as other NAD^+^ precursors, triggers autophagy through the activation of sirtuins [37] and metformin though the activation of AMPK [38]. SMER-28 is a small-molecule that enhances autophagy via an ATG5-dependent pathway [39], whereas trehalose, raffinose, and sucrose are saccharides that can increase autophagy in an ATG5-dependent manner [19,40,41]. The obtained results evidenced an increase in LC3-II levels following the administration of trehalose and raffinose as compared with all the other treatments. Both trehalose and raffinose did not show any cytotoxic effect, as demonstrated by others [42,43]. Among these, trehalose induced an increase of expression of other autophagy markers, i.e., Beclin1 and ATG5, as already documented [40]. Considering these results, we focused our attention on trehalose, investigating its ability to induce autophagy and counteract gliadin peptides cytotoxicity. In fact, trehalose is a non-toxic disaccharide found in bacteria, yeasts, fungi, plants, and invertebrates used as sources of energy and carbon. In some organisms this sugar acts as a signaling molecule and protects cells against a variety of environmental stresses (e.g., dehydration, heat, cold, and oxidation) [44]. In addition, trehalose showed anti-aggregating and molecular chaperone properties in models of proteopathies such as Huntington and Parkinson diseases [21,45] as well as anti-oxidative ones through the activation of the p62-Keap1/Nrf2 pathway [20]. To study the effect of trehalose in presence of digested gliadin, full confluent Caco-2 cells monolayer was treated with both digested gliadin and the investigated disaccharide. As observed by cytofluorimetric and immunoblotting assays, trehalose increased LC3-II levels and rescued the autophagy flux in presence of digested gliadin. Moreover, the beneficial effect exerted by trehalose in presence of gliadin was highlighted by the increase in cell viability. Furthermore, the ability of this disaccharide to induce autophagy in presence/absence of digested gliadin was confirmed and strengthened by the MIFC study, in particular through spot-count analysis. Then, cytofluorimetric analysis showed that trehalose reduced the intracellular amount of digested fluorescent gliadin. These results can be explained by taking into consideration the anti-aggregating and anti-oxidative properties of trehalose, which can sustain the cellular response to oxidative stresses and promote gliadin degradation. Similar experiments were finally performed on confluent HT-29 cells in order to determine if digested gliadin could exert toxic effects similar to those observed in the Caco-2 cell model. Again, digested gliadin administration decreased cell viability as compared with digested casein and an autophagy blockage was detected after the cytofluorimetric analysis of LC3-II protein levels. Furthermore, the autophagy flux was rescued following trehalose administration in presence of digested gliadin. 

Altogether, these results confirm that autophagy is implicated in the cellular response triggered by enzymatically digested gliadin peptides. Moreover, we reported that autophagy induction through trehalose rescues viability leading to the degradation of intracellular digested gliadin, underlining that autophagy modulation could be a promising approach in counteracting gluten cytotoxicity. Considering this evidence, a parsimonious model of interpretation is proposed (Figure 15).

As shown, enzymatically digested gliadin spontaneous internalization [33] leads to an accumulation of these toxic peptides in endocytic vesicles resulting in an increase of oxidative stress [6]. As previously established, the presence of ROS triggers an inflammatory response and autophagy plays a crucial role in the regulation of this process. Specifically, autophagy regulates inflammatory activation through the degradation of inflammasomes and regulation of pro-inflammatory cytokines release, and its impairment is associated with dysregulated inflammatory responses [46,47]. Down-regulation of the key autophagy-regulatory gene BECN1 recently observed in the biopsies of pediatric CD patients [13], suggests that autophagy is impaired in CD patients as a consequence of a long exposure to gliadin peptides. Autophagy blockage, as already mentioned, leads to cell death mediated by apoptosis thus contributing to villous atrophy. 

Altogether, these findings show that digested gliadin administration causes autophagy blockage in an in vitro model based on full confluent Caco-2 and HT-29 cell lines. Moreover, autophagy induction modulated by trehalose rescues cell viability as a consequence of intracellular gliadin digestion, suggesting that this disaccharide could exert a positive role in counteracting gliadin cytotoxicity.

Future experiments will be directed to characterize an autophagy response in a fully differentiated Caco-2 cell in vitro model in order to investigate whether autophagy induction could also restore the permeability barrier function and the correct expression of TJs proteins. Moreover, the role of transglutaminase 2 (TGM2) will be assayed because of its important role in controlling the balance between autophagy and apoptosis [48].

## Figures and Tables

**Figure 1 cells-08-00348-f001:**
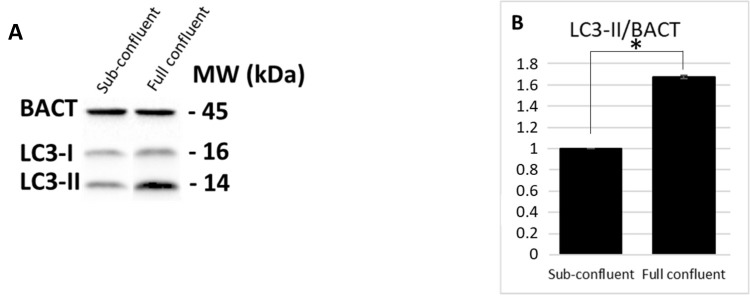
LC3-II expression levels of Caco-2 cells at different confluences. LC3-II and BACT protein expression were analyzed through immunoblotting (**A**) and densitometric analyses (**B**). LC3-II was normalized to BACT levels as recommended [23]. Normalized values are reported on the y-axis as arbitrary units. Molecular weights (kDa) and standard error (SE) bars are reported. The asterisks indicate *p* < 0.05, one-way Anova. The experiments were performed in duplicate.

**Figure 2 cells-08-00348-f002:**
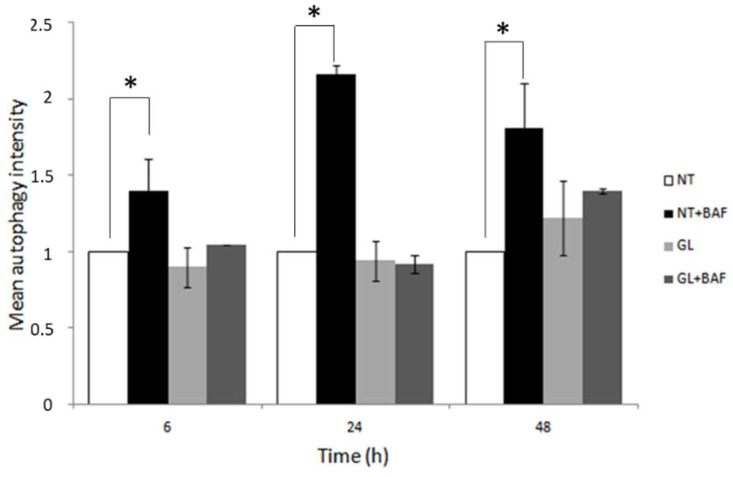
LC3-II expression levels in full confluent Caco-2 cells monolayer after PT-gliadin administration (GL). The LC3-II levels in Caco-2 cells cultured 5 days after confluence and treated with PT-gliadin (1 µg/µL) and bafilomycin A1 (10 nM) (BAF). Measurements were performed using a Muse^®^ Cell Analyzer (Merck) at different times. Results were normalized on the non-treated (NT) samples. SE bars are reported. The asterisks indicates *p* < 0.05, Anova one-way, as compared with NT samples. The experiments were performed in triplicate. Cytofluorimetric plots are reported in Appendix A (Appendix A).

**Figure 3 cells-08-00348-f003:**
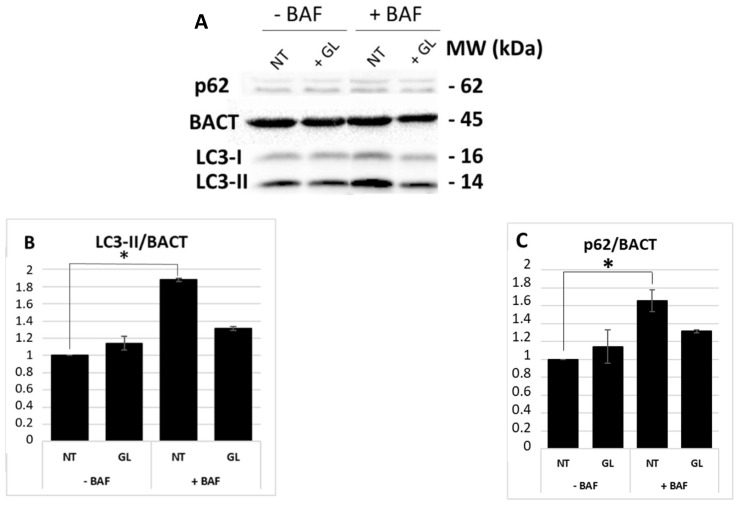
LC3-II and p62 expression levels at 24 h p.t. after PT-gliadin administration. LC3-II, p62, and BACT protein expression were analyzed through immunoblotting (**A**) and densitometric analyses (**B**,**C**). LC3-II and p62 were normalized to BACT levels as recommended [23]. Normalized values are reported on the y-axis as arbitrary units. Molecular weights (kDa) and SE bars are reported. The asterisks indicate *p* < 0.05, Anova one-way, as compared with relative controls. The experiments were performed in triplicate.

**Figure 4 cells-08-00348-f004:**
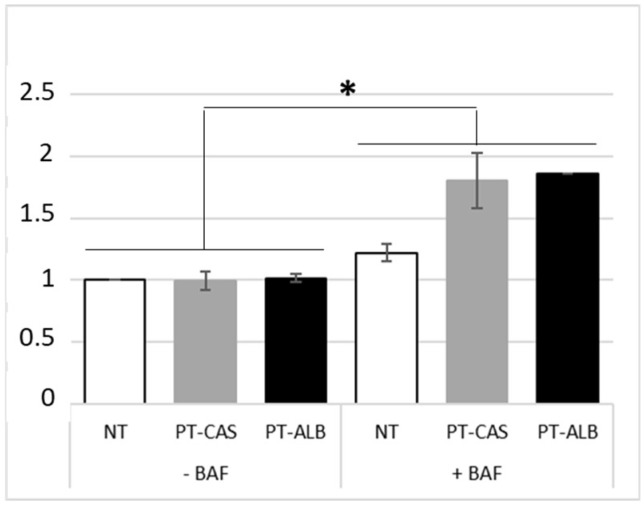
LC3-II expression levels in Caco-2 cells monolayer after administration of enzymatically digested casein and albumin (each 1 µg/µL) in presence/absence of bafilomycin A1 (10 nM). Measurements were performed using a Muse^®^ Cell Analyzer (Merck) at 24 h p.t. Results were normalized on the non-treated (NT) sample. SE bars are reported. The asterisks indicates *p* < 0.05, Anova one-way, as compared with both NT samples and relative controls. The experiments were performed in duplicate. Cytofluorimetric plots are reported in Appendix A (Appendix A).

**Figure 5 cells-08-00348-f005:**
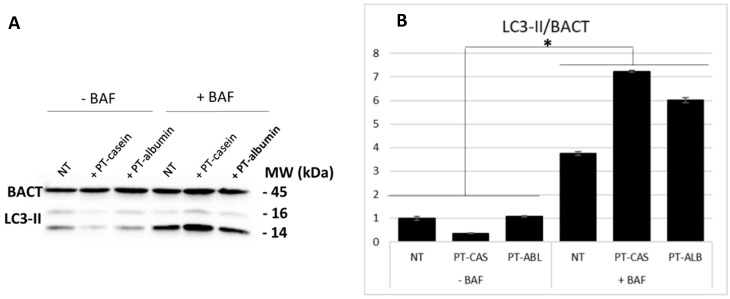
LC3-II levels at 24 h p.t. after PT-casein and PT-albumin administration. LC3-II protein expression was analyzed through immunoblotting (**A**) and densitometric analyses (**B**). LC3-II was normalized to BACT levels. Normalized values are reported on the y-axis as arbitrary units. Molecular weights (kDa) and SE bars are reported. The asterisks indicate *p* < 0.05, Anova one-way, as compared with relative controls and NT sample. The experiments were performed in duplicate.

**Figure 6 cells-08-00348-f006:**
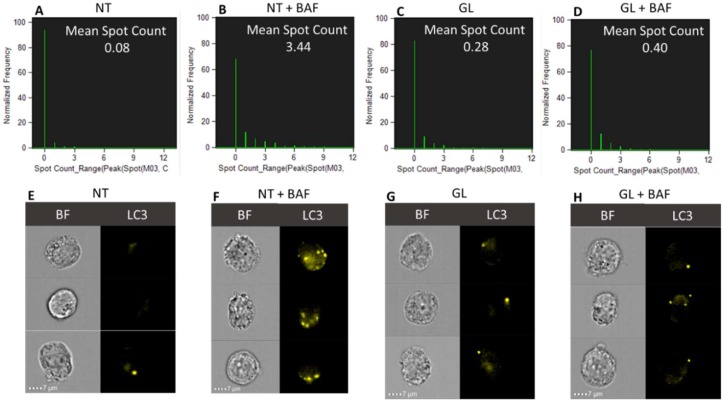
LC3-II spot-count histograms and cells galleries. The spot-count feature Spot Count_Range(Peak(Spot(M03,Ch3-LC3-AF555,Bright,7,3),Ch3,Bright,0) was used for NT (**A**), NT + BAF (**B**), GL (**C**), and GL + BAF (**D**). The mean spot counts for NT, NT + BAF, GL, and GL + BAF are 0.08, 3.44, 0.28, and 0.40, respectively. Brightfield (BF) and LC3-AF555 (yellow) images of the cells are shown for NT (**E**), NT + BAF (**F**), GL (**G**), and GL + BAF (**H**). Event collected: 2000. The analysis was performed using Amnis ImageStream X Mark II (Merck).

**Figure 7 cells-08-00348-f007:**
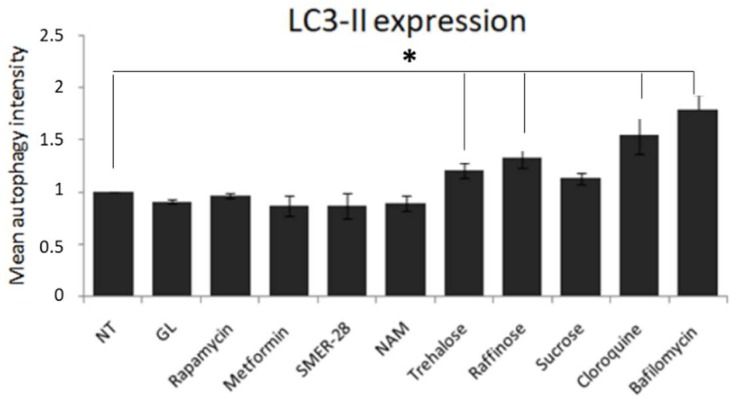
Cytofluorimetric analysis of LC3-II levels in Caco-2 confluent cells treated with different autophagy inducers. Rapamycin (5 µM), metformin (5 mM), SMER-28 (50 µM), NAM (5 mM), trehalose (100 mM), raffinose (100 mM), sucrose (100 mM), chloroquine (25 µM), and bafilomycin A1 (10 nM) were used to assess the autophagic flux. LC3-II expression values were normalized to NT sample. The analysis was performed using a Muse^®^ Cell Analyzer (Merck). The asterisks indicate *p* < 0.05, ANOVA one-way, as compared with the NT sample. SE bars are reported. The experiments were performed in triplicate. Cytofluorimetric plot are reported in Appendix A (Appendix A).

**Figure 8 cells-08-00348-f008:**
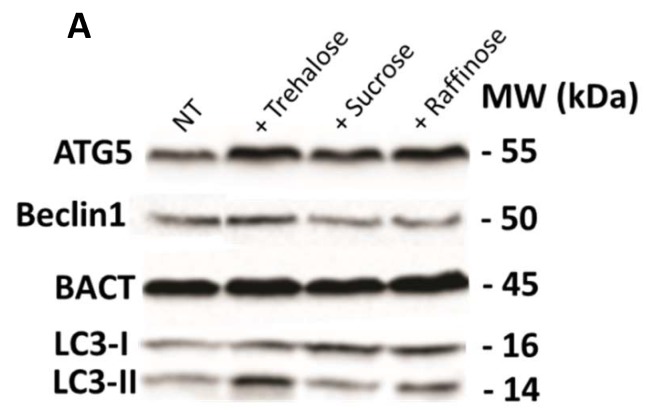
Immunoblotting of ATG5, Beclin1, and LC3-II protein levels (**A**) and relative densitometric analysis (**B**). Caco-2 confluent cells were treated with trehalose, raffinose, and sucrose (100 mM) and collected after 24 h p.t. Protein expression levels were normalized to BACT levels. Normalized values are reported on the y-axis as arbitrary units. Molecular weights (kDa) are reported. The asterisks indicate *p* < 0.05, ANOVA one-way, as compared with the NT sample. SE bars are reported. Experiments were performed in duplicate.

**Figure 9 cells-08-00348-f009:**
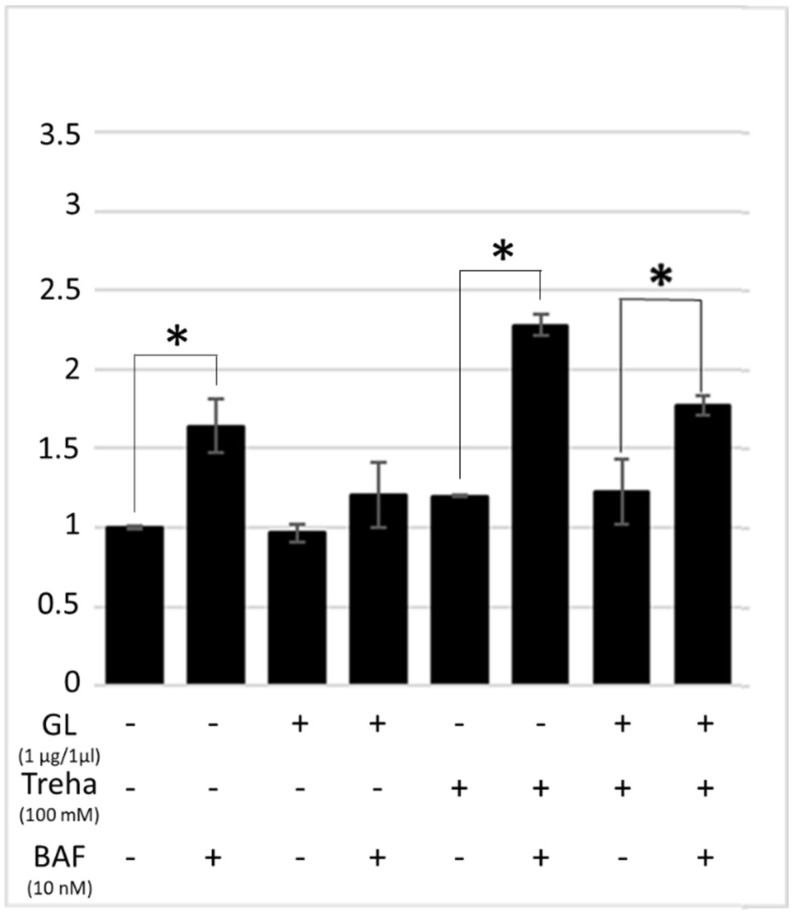
Cytofluorimetric analysis of LC3-II levels in Caco-2 cells treated with PT-gliadin (1 µg/µL) in presence/absence of trehalose (Treha) or raffinose (100 mM). Bafilomycin A1 (10 nM) was used to monitor differences the autophagic flux. The LC3-II expression values were normalized on the NT sample. The analysis was performed using a Muse^®^ Cell Analyzer (Merck). The asterisks indicate *p* < 0.05, ANOVA one-way, as compared with the relative control samples. SE bars are reported. The experiments were performed in triplicate. Cytofluorimetric plot are reported in Appendix A (Appendix A).

**Figure 10 cells-08-00348-f010:**
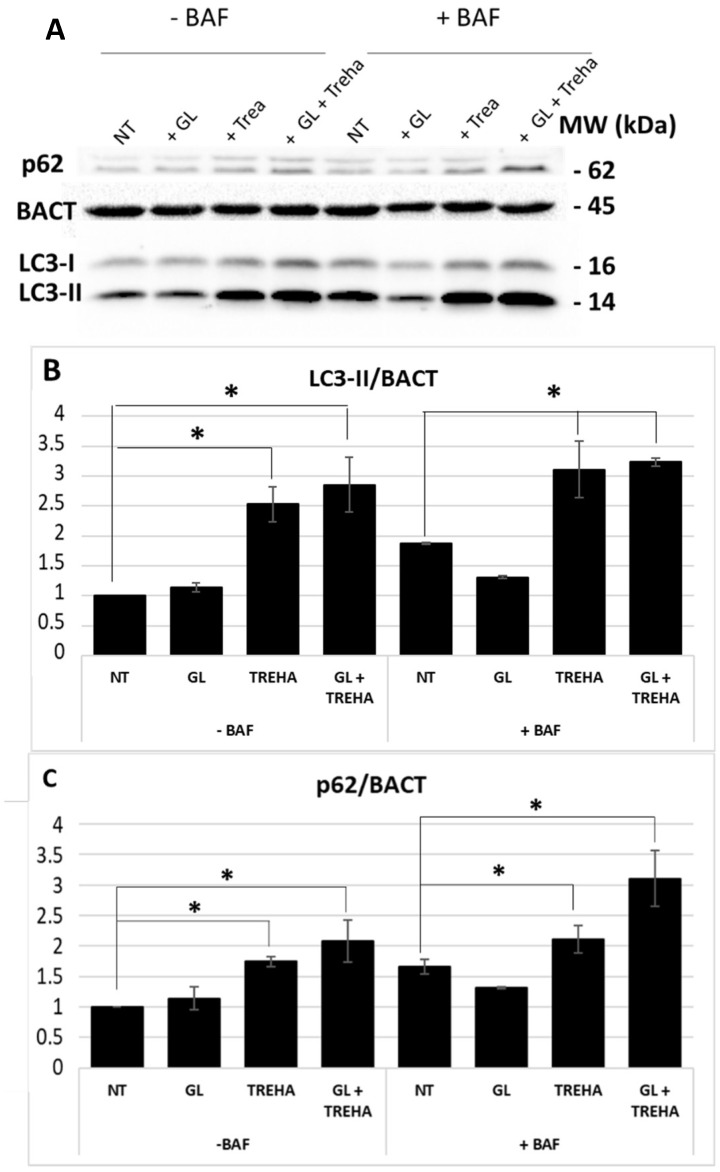
LC3-II and p62 proteins expression at 24 h p.t. after PT-gliadin (1 µg/µL) and trehalose (100 mM) administration. The LC3-II and p62 expressions were analyzed through immunoblotting (**A**) and densitometric analyses (**B**,**C**) and normalized to BACT levels [23]. Normalized values are reported on the y-axis as arbitrary units. Molecular weights (kDa) and SE bars are reported. The asterisks indicate *p* < 0.05, Anova one-way, as compared with NT samples. The experiments were performed in triplicate.

**Figure 11 cells-08-00348-f011:**
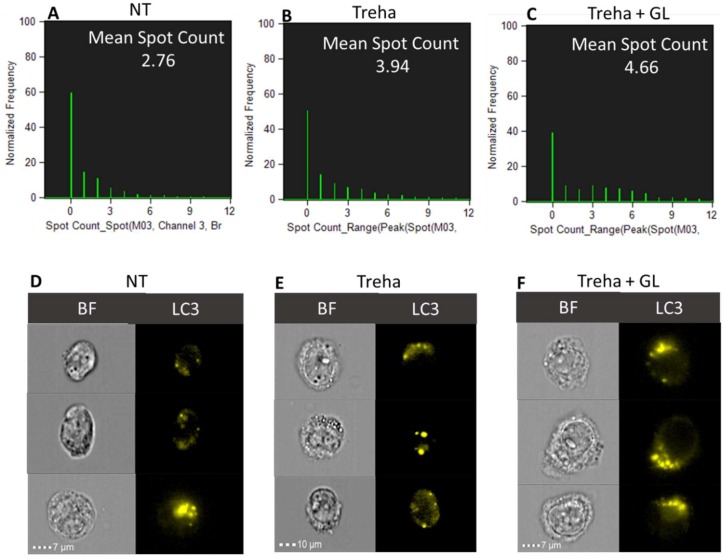
LC3 Spot-count histograms and cells galleries. The spot-count feature Spot Count_Range(Peak(Spot(M03,Ch3-LC3-AF555,Bright,7,3),Ch3,Bright,0) was used for NT (**A**), Treha (**B**), and Treha + GL (**C**). The mean spot counts for NT, Treha, and Treha + GL are 2.76, 3.94, and 4.66, respectively. Brightfield (BF) and LC3-AF555 (yellow) images of the cells are shown for NT (**D**), Treha (**E**), and Treha + GL (**F**). Event collected: 2000. The analysis was performed using Amnis ImageStream X Mark II (Merck).

**Figure 12 cells-08-00348-f012:**
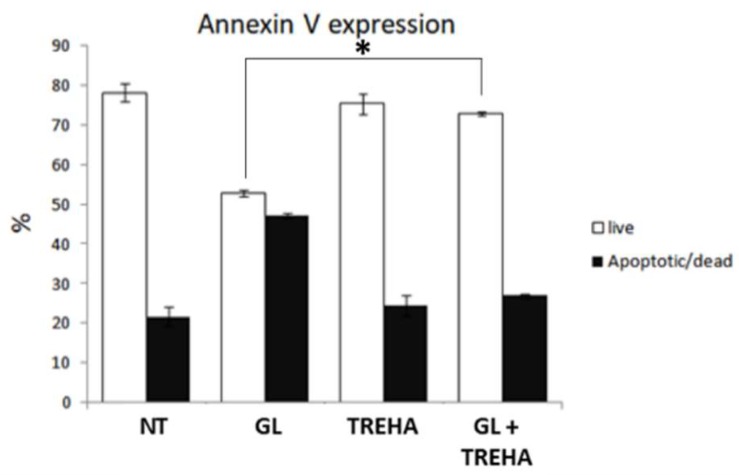
Viability and apoptotic analysis of Caco-2 cells after treatment with digested gliadin (1 µg/µL), trehalose (100 mM), and the combination of both. The Annexin V protocol was performed using a Muse^®^ Cell Analyzer (Merck) at 24 h p.t. SE bars are reported. The asterisk indicates *p* < 0.05, ANOVA one-way, as compared with NT sample. The experiments were performed in triplicate. Cytofluorimetric plots are reported in Figure 8.

**Figure 13 cells-08-00348-f013:**
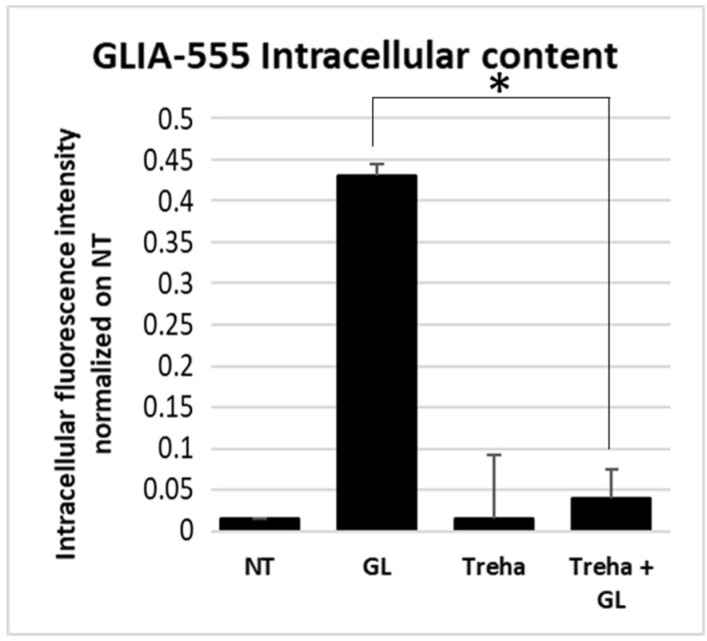
Analysis of the intracellular amount of fluorescent PT-gliadin (GLIA-555) in presence/absence of trehalose. Caco-2 cells were treated with GLIA-555 (1 µg/µL), trehalose (100 mM), and the combination of both. Cytofluorimetric analysis was performed using a Muse^®^ Cell Analyzer (Merck) after 24 h p.t. The asterisk indicates *p* < 0.05, ANOVA one-way. SE bars are reported. The experiments were performed in triplicate. Cytofluorimetric plots are reported in Figure 9.

**Figure 14 cells-08-00348-f014:**
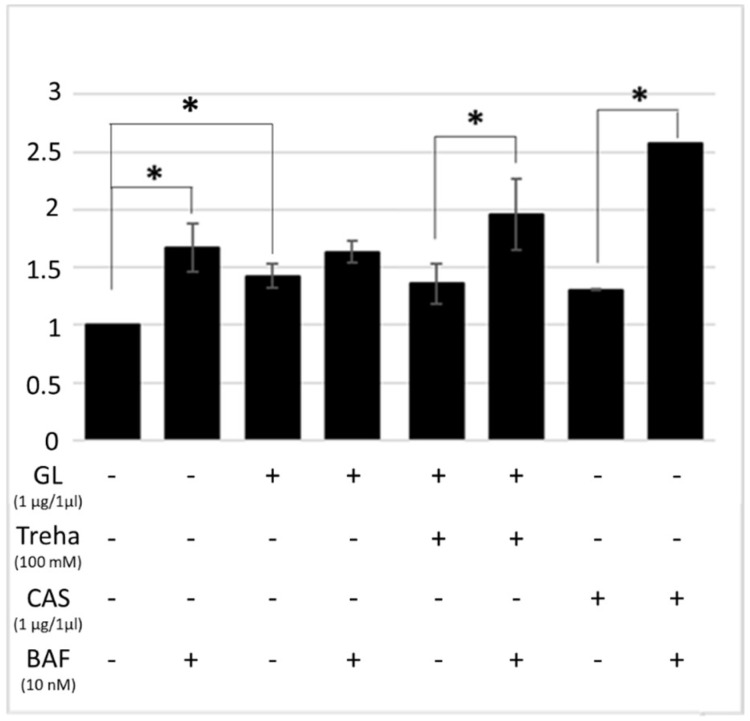
Cytofluorimetric analysis of LC3-II levels in HT-29 cells treated with digested gliadin (1 µg/µL) in presence/absence of trehalose (100 mM). Bafilomycin A1 (10 nM) was used to monitor the autophagic flux. The LC3-II expression values were normalized on the NT sample. The analysis was performed using a Muse^®^ Cell Analyzer (Merck). The asterisks indicate *p* < 0.05, ANOVA one-way, as compared with the relative control samples. SE bars are reported. The experiments were performed in triplicate. Cytofluorimetric plot are reported in Figure 11.

**Figure 15 cells-08-00348-f015:**
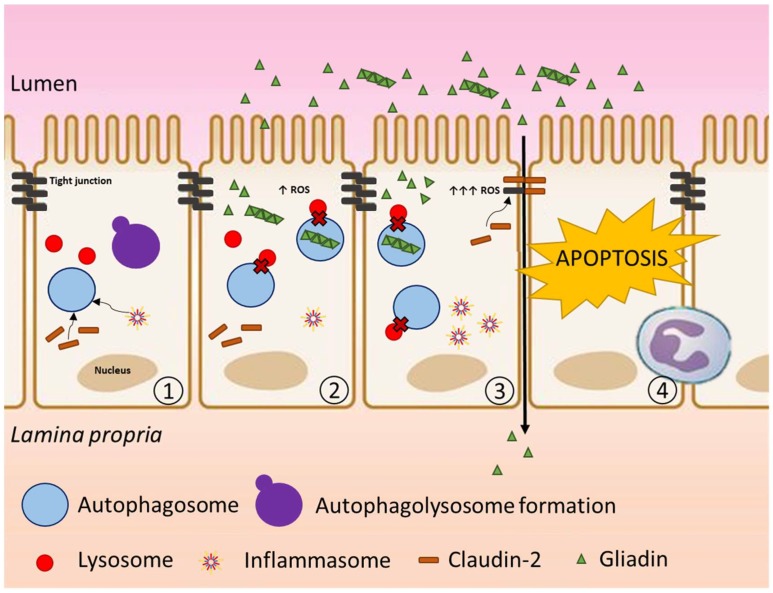
Interpretative model of autophagy involvement in CD. (1) Normal enterocyte: autophagy flux is active and is involved in the control of the cellular homeostasis, e.g., degradation of claudin-2 and inflammasome negative regulation. (2) Gliadin peptides enter in the intestinal epithelial cell generating ROS and triggering an autophagy response. Autophagy blockage occurs as a consequence of the stress persistence. (3) Because of the autophagy blockage, oxidative stress increased thus contributing to a dysregulated inflammatory response caused by the lack of degradation inflammasomes. Tight junctions (TJs) formation is compromised leading to an increase in paracellular permeability. (4) Inflammation and intraepithelial lymphocytes (IELs) activation lead to apoptosis of the intestinal epithelial cells.

**Table 1 cells-08-00348-t001:** Schematic representation of the experimental plan and the techniques adopted in each step. IB: immunoblotting; FC: flow cytometry; MIFC: multispectral imaging flow cytometry.

Stages	Description	Techniques	Figures
1	Effects of digest gliadin on Caco-2	IB, FC, MIFC	1,2,3,4,5,6
2	Evaluation of different autophagy inducers	FC	7
3	Identification of trehalose as the best candidate	IB	8
4	Beneficial effects of trehalose against digested gliadin	IB, FC, MIFC	9,10,11,12,13
5	Comparison with HT-29 cells	FC	14

**Table 2 cells-08-00348-t002:** Cell viability and apoptotic rate of Caco-2 cells treated with trehalose and raffinose (100 mM). The Caco-2 cells monolayer was treated with saccharides for 24 h and then analyzed through Annexin V cytofluorimetric analysis using a Muse^®^ Cell Analyzer (Merck). The percentage values obtained are reported as mean ± SE. The experiments were performed in triplicate. Cytofluorimetric plots are reported in Appendix A (Appendix A).

	Live	Apoptotic/dead
**NT**	78.17 ± 2.28	21.82 ± 2.28
**Trehalose**	75.40 ± 2.45	24.60 ± 2.45
**Raffinose**	71.97 ± 4.43	28.02 ± 4.43

**Table 3 cells-08-00348-t003:** Apoptotic analysis of HT-29 cells after treatment with digested gliadin or casein (1 µg/µL). The Annexin V protocol was performed using a Muse^®^ Cell Analyzer (Merck) at 24 h p.t. The obtained values are reported as mean ± SE. The asterisks indicate *p* < 0.05, ANOVA one-way, as compared with NT sample. The experiments were performed in triplicate. Cytofluorimetric plots are reported in Figure 10.

	Live	Apoptotic/dead
**NT**	88.49 ± 3.18	11.51 ± 3.18
**PT-gliadin**	69.57 ± 5.27 *****	30.43 ± 5.27 *****
**PT-casein**	85.11 ± 2.57	14.89 ± 2.57

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
