# Peer review of "Trehalose Modulates Autophagy Process to Counteract Gliadin Cytotoxicity in an In Vitro Celiac Disease Model"

_cells, 2019, doi:10.3390/cells8040348_

Round 1
Reviewer 1 Report
The paper by Manai et al describes the ability of trehalose, a naturally occurring disaccharide, and an mTOR –indipendent autophagy inducer, to rescue the autophagy flux in caco-2 and HT-29 cells exposed to pt gliadin resulting in an improvement of cell viability. Trehalose Increases autophagic flux and counteracts
PT-Gliadin cytotoxicity also by reducing the Intracellular entrance of these peptides in caco-2 cells.
The observed autophagy blockage exerted by PT gliadin is explained considering the ability of the immunodominant 33-mer peptide to spontaneously form supramolecular structures, whose degradation is difficult, and the increase of ROS caused by the internalization and the accumulation of the p31-43 peptide in the lysosomes. Autophagy plays in the maintenance of intestinal homeostasis and autophagy induction seems to be a promising approach to counteract the toxic effects of gliadin peptides.
Specific comments
Line 40 “Gluten, a composite of storage proteins, present in wheat, barley, rye oats, and related species”. The verb is missing
Line 42. “gliadin, which shows toxic properties also after the enzymatic digestion mediated by pepsin and trypsin” . Gliadin exerts toxic properties not also, but especially after the enzymatic digestion by pepsin and trypsin, since in crude gliadin, some epitopes can be masked and not available for immune recognition.
Line 49-50 There are few evidence supporting the hypothesis that autophagy could play a role in CD pathogenesis. It could be mentioned that few evidences are related to previous studies ref 12 and 13 of the authors.
Line 61 as well as prevent the aggregation of toxic peptides , yes, but the reference 21 is related to Huntington's disease models., should be reported
Where is reference 22 in the text? All the reference must be renumbered
Line 81, reference 23 is not relative to the gliadin digestion method but the authors of the reference adopted it too
Line 88 reference by Manai et al. 2018 should be numbered
Line 96 reference 49 is not adequate as the number and as a method for immunoblotting
Results
All the western blots proposed in the paper show molecular weight (KD) of proteins not in order of migration, this is a very strange manner to represent it. Furthermore, all the densitometric analysis do not have a unit of measure.
Fig 1 determination of autophagy level ( expression of LC3-II) in subconfluent versus full confluent Caco-2 cells has been yet described by Tuncer et al, 2019 ( ref 27 of the paper).
Line 144 to mimic the gastrointestinal digestion according to [23] ..
please mention the author and not only the number or insert as previously described
Line 145… and administered at the concentration of 1 μg/μl to Caco-2 cells and cultured for 5 days after they reached complete monolayer confluence . Does the treatment with Pt gliadin last for 5 days??
Fig 2 and 3 NT + BAF are cells treated with BAF, so eliminate NT for this group.. No differences were found in LC3-II and p62 expression between Caco-2 treated with PT-glia and Caco-2 not treated (NT), but how is it possible that in the previous paper published by the same authors, ref 12 Manai et al, 2018 this difference was detected? It’s really confusing.
Fig.6 Apoptotic analysis of caco-2 cells after treatment with PT gliadin has been yet described by Giovannini et al. , 2000 and by the same authors of this paper This figure is not necessary for this manuscript.
Fig.9 Immunoblotting of ATG5, Beclin1 and LC3-II level in caco-2 cells treated with trehalose, sucrose, and raffinose. Why the authors do indicate the expression levels normalized to NT sample by numbers and not by densitometric analysis as the other blots of the paper?
Line 215: as previously documented .. please specify the reference
Author Response
We have received your comments and we are grateful for your suggestions and keen observations. We have provided a point-by-point response in the attached Word file. The manuscript was corrected according to the Reviewer's specific comments.

Reviewer 2 Report
The research presented in the article could be interesting, but the manuscript requires a significant improvement.
- The title raises doubts because it suggests that the authors have conducted an experiment on the in vitro celiac disease model. However, the current study was carried out on Caco-2 cells. These cells derived from human colon adenocarcinoma are capable of enterocytic-like differentiation ect. and therefore, are commonly accepted as the in vitro model for bioavailability assessments.
- Abstract: line 23-26 - I think it is unnecessary in this place. Abstract is not a place for quoting the results of previous research. It should indicate the purpose of the present study, the main results and conclusions
- The introduction need to explain and give reasonable references to prove and clarify on what basis Caco-2 cells model was recognized as in vitro model of celiac disease. Could it be as simple? Caco-2 are the model of intestinal epithelial cell, but maybe new research appeared allowing to suggest Caco-2 as in vitro model of CD.
- The hypothesis of research appears in the discussion. I suggest you move it to the introduction,
- In general, the presented results is very extensive (and laborious) but the methodological presentation of the research do not allow to track the experiment, therefore I suggest to add a scheme that would accurately represent the subsequent stages of the experiment,
- There is no separate subsection on statistical analysis
- Results chapter: avoid unnecessary methodological information that should appear in the Matarials and Methods section (for example inlines 143-148; 171-175; 208-210…)
- Again, the study is very extensive therefore try to focus on the most important findings, other results can be move to supplementary materials
- There is no conclusions or summery, instead the Authors present their research plans for the future
Minor comments:
- Explain the abbreviations,
- Add details (city, country ) of every producers
- refer to Figures and Tables in the main text;
- Fig 1 and Fig 5: add “A” for identification of immunoblotting analysis and “B” – for densitometric; besides explain what the Y-axis presents
Author Response

(The authors gave the same response as above.)

Round 2
Reviewer 1 Report
The authors provided a point-by-point response, the paper has improved a lot.
Reviewer 2 Report
The reviewed manuscript has been significantly improved,my doubts have been clarified. I accept all responses to my suggestions. I suggest that in this form the manuscript may be published